# Hierarchical Preference Optimization: Learning to achieve goals via feasible subgoals prediction

## Abstract

This work introduces Hierarchical Preference Optimization (HPO), a novel approach to hierarchical reinforcement learning (HRL) that addresses non-stationarity and infeasible subgoal generation issues when solving complex robotic control tasks. HPO leverages maximum entropy reinforcement learning combined with token-level Direct Preference Optimization (DPO), eliminating the need for pre-trained reference policies that are typically unavailable in challenging robotic scenarios. Mathematically, we formulate HRL as a bi-level optimization problem and transform it into a primitive-regularized DPO formulation, ensuring feasible subgoal generation and avoiding degenerate solutions. Extensive experiments on challenging robotic navigation and manipulation tasks demonstrate HPO's impressive performance, where HPO shows an improvement of upto 35% over the baselines. Furthermore, ablation studies validate our design choices, and quantitative analyses confirm HPO's ability to mitigate non-stationarity and infeasible subgoal generation issues in HRL.

## 1 Introduction

Reinforcement learning (RL) encounters significant challenges in sparse-reward environments, particularly in complex robotic control tasks (Nair et al., 2018). Hierarchical Reinforcement Learning (HRL) (Sutton et al., 1999) helps to deal with sparse reward issues by enhancing exploration and introducing temporal abstraction (Nachum et al., 2019). However, off-policy HRL methods (Levy et al., 2018; Nachum et al., 2018) face their own challenges. The first (**C1**) is non-stationarity caused by evolving lower-level policies, which destabilizes the higher-level reward function (Chane-Sane et al., 2021). The second (**C2**) is the high-level policy's tendency to generate subgoals that are infeasible for the lower-level policy to achieve.

Preference-based learning (PbL) approaches like Reinforcement Learning from Human Feedback (RLHF) (Christiano et al., 2017) have been successful in solving complex tasks. However, integrating PbL directly with HRL is non-trivial. Singh et al. (2024) proposed an RLHF-based method to mitigate non-stationarity in HRL by first learning a reward model from preference data and then using RL to train the high-level policy. However, the approach in Singh et al. (2024) introduces an expensive and potentially unstable RL step which can still result in infeasible subgoal predictions for the lower-level primitives.

Recently, direct preference optimization (DPO) methods (Rafailov et al., 2024b) have been introduced, which learn policies directly from preference data, bypassing the need for RL. Leveraging DPO to learn the high-level policy in HRL could efficiently mitigate non-stationarity. However, naively applying DPO requires effective pre-trained reference policies, which are often unavailable in robotics scenarios. Additionally, such approaches may still produce subgoals that are infeasible for the lower-level policy to execute.

In this work, we propose Hierarchical Preference Optimization (HPO), a hierarchical approach that optimizes the higher level policy using token-level DPO objective (Rafailov et al., 2024a), and the lower level primitive policy using RL. Since HPO learns higher level policy from preference data, it avoids dependence on changing lower level primitive and thus mitigates non-stationarity (C1). We derive token-level DPO objective in a maximum entropy RL setting, to eliminate the requirement

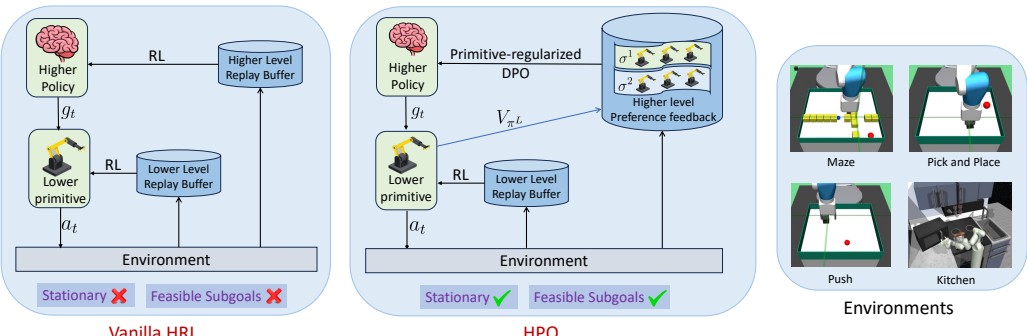

Figure 1: `HPO` Overview: (left) In vanilla HRL, the higher level rewards depend on the environment rewards, and thus on the lower primitive behavior, which causes non-stationarity in HRL. Also, the higher level may predict infeasible subgoals that are too hard for lower primitive. (middle) In `HPO`, the lower level value function $V_{\pi^L}$ is leveraged to condition higher level policy into predicting feasible subgoals, and direct preference optimization (`DPO`) is used to optimize higher level policy. Since this preference-based learning approach does not depend on lower primitive, this mitigates non-stationarity. Note that since the *current* estimation of value function is used to regularize the higher policy, it does not cause non-stationarity. (right) Training environments: $(i)$ maze navigation, $(ii)$ pick and place, $(iii)$ push, and $(iv)$ franka kitchen environment.

of pre-trained models as reference policy. Furthermore, in order to generate feasible subgoals, we formulate HRL as a bi-level formulation, which regularizes the higher level policy to predict feasible subgoals using lower level value function based regularization, which we call *primitive regularization* (thus mitigating C2). Finally, since eliciting human feedback is hard in complex robotic tasks, we employ primitive-in-the-loop approach (Singh et al., 2024) for autonomously generating preferences using sparse environment rewards. The primary contributions of this work are as follows.

**(i) Novel primitive-regularized preference optimization approach to solve hierarchal RL:** We propose a novel Hierarchical Preference Optimization (`HPO`) method that leverages primitive-regularized Direct Preference Optimization (`DPO`) to solve complex RL tasks using human preference data (Section 4). Our approach is principled; we derive it by reformulating the HRL problem as a bi-level optimization problem. To the best of our knowledge, this is the first work to utilize the bi-level optimization framework to develop a principled solution for HRL.

**(ii) Addressing non-stationarity and infeasible subgoal generation issues of SoTA HRL:** By adopting a principled approach, we solve the bi-level optimization problem while respecting the distributional dependence between lower-level and higher-level policies. This enables our approach to significantly reduce the effects of non-stationarity and infeasible subgoal generation in a variety of scenarios. We provide detailed evidence of this in Section 5 Figures 3 and 4.

**(iii) Improved performance in complex robotics tasks:** Our extensive experiments demonstrate that `HPO` demonstrates an improvement of upto $35\%$ over the baselines in most of the task environments, where other baselines typically fail to make any significant progress (Section 5).

## 2 RELATED WORKS

**Hierarchical Reinforcement Learning.** `HRL` is an elegant framework that promises the intuitive benefits of temporal abstraction and improved exploration (Nachum et al., 2019). Prior research work has focused on developing efficient methods that leverage hierarchical learning to efficiently solve complex tasks (Sutton et al., 1999; Barto and Mahadevan, 2003; Parr and Russell, 1998; Dietterich, 1999). Goal-conditioned `HRL` is an important framework in which a higher-level policy assigns subgoals to a lower-level policy (Dayan and Hinton, 1992; Vezhnevets et al., 2017), which executes primitive actions on the environment. Despite its advantages, `HRL` faces challenges owing to its hierarchical structure, as goal-conditioned `RL` based approaches suffer from non-stationarity in off-policy settings where multiple levels are trained concurrently (Nachum et al., 2018; Levy et al., 2018). These issues arise because the lower level policy behavior is sub-optimal and unstable. Prior works deal with these issues by either simulating optimal lower primitive behavior (Levy et al.,

2018), relabeling replay buffer transitions (Nachum et al., 2018; Singh and Namboodiri, 2023b;a), or assuming access to privileged information like expert demonstrations or preferences (Singh et al., 2024; Singh and Namboodiri, 2023a;b). In contrast, we propose a novel bi-level formulation to mitigate non-stationarity and regularize the higher-level policy to generate feasible subgoals for the lower-level policy.

**Behavior Priors.** Some prior work relies on hand-crafted actions or behavior priors to accelerate learning (Nasiriany et al., 2021; Dalal et al., 2021). While these methods can simplify hierarchical learning, their performance heavily depends on the quality of the priors; sub-optimal priors may lead to sub-optimal performance. In contrast, ours is an end-to-end approach that does not require prior specification, thereby avoiding significant expert human effort.

**Preference-based Learning.** A variety of methods have been developed in this area to apply reinforcement learning (RL) to human preference data (Knox and Stone, 2009; Pilarski et al., 2011; Wilson et al., 2012; Daniel et al., 2015), that typically involve collecting preference data from human annotators, which is then used to guide downstream learning. Prior works in this area (Christiano et al., 2017; Lee et al., 2021) first train a reward model based on the preference data, and subsequently employ `RL` to derive an optimal policy for that reward model. More recent approaches have focused on improving sample efficiency using off-policy policy gradient methods (Haarnoja et al., 2018) to learn the policy. Recently, direct preference optimization based approaches have emerged (Rafailov et al., 2024b;a; Hejna et al., 2023), which bypass the need to learn a reward model and subsequent RL step, by directly optimizing the policy with a KL-regularized maximum likelihood objective corresponding to a pre-trained model. In this work, we build on the foundational knowledge in maximum entropy `RL` (Ziebart, 2010), and derive a token-level direct preference optimization (Rafailov et al., 2024b;a) objective regularized by lower-level primitive, resulting in an efficient hierarchical framework capable of solving complex robotic tasks.

## 3 PROBLEM FORMULATION

**MDP Setup.** We formulate the problem using a Markov Decision Process (MDP), defined as $(\mathcal{S}, \mathcal{A}, p, r, \gamma)$. Here, $\mathcal{S}$ represents the state space, and $\mathcal{A}$ denotes the action space. The transition dynamics are governed by the stochastic probability function $p : \mathcal{S} \times \mathcal{A} \to \Delta(\mathcal{S})$, which maps each state-action pair to a probability distribution over subsequent states. The reward function, $r : \mathcal{S} \times \mathcal{A} \to \mathbb{R}$, provides a scalar reward based on the current state and action, and $\gamma \in (0, 1)$ is the discount factor that modulates the importance of future rewards. At each time step $t$, the agent follows a policy $\pi : \mathcal{S} \to \Delta(\mathcal{A})$, which defines a probability distribution over actions $a_t$ given the current state $s_t$. After taking action $a_t$, the agent receives a reward $r_t = r(s_t, a_t)$, and the environment transitions to a new state $s_{t+1}$ according to the transition probability $p(\cdot|s_t, a_t)$.

**RL Objective.** The objective in RL is to find an optimal policy that maximizes the expected cumulative reward, formally defined as $\pi^* := \arg\max_\pi J(\pi)$, where $J(\pi) = \mathbb{E}_\pi \left[ \sum_{t=0}^\infty \gamma^t r_t \right]$. The value function for policy $\pi$, denoted as $V_\pi(s_t, g_t)$, captures the expected cumulative reward from state $s_t$ with goal $g_t$, and is defined as $V_\pi(s_t, g_t) = \mathbb{E}_\pi[\sum_{t=0}^T \gamma^t r_t | s_t, g_t]$, where the expectation is over policy trajectories, $\gamma$ is the discount factor, and $r_t$ is the reward at time $t$.

**Goal-Conditioned `HRL`.** In this framework, the higher-level policy generates subgoals for the lower-level policy, which then takes primitive actions to achieve these subgoals, collectively working toward the overall objective. Formally, the higher-level policy $\pi^H : \mathcal{S} \to \Delta(\mathcal{G})$ selects a subgoal $g_t \in \mathcal{G}$, where $\mathcal{G}$ is a subset of the state space $\mathcal{S}$, representing the set of goals. A subgoal $g_t \sim \pi^H(\cdot|s_t)$ is selected every $k$ timesteps, such that $g_t = g_{k \cdot \lceil t/k \rceil}$ and remains fixed during this period. At each timestep $t$, the lower-level policy $\pi^L : \mathcal{S} \times \mathcal{G} \to \Delta(\mathcal{A})$ selects an action $a_t \sim \pi^L(\cdot|s_t, g_t)$ based on the current state $s_t$ and the subgoal $g_t$, after which the environment transitions to the next state, $s_{t+1} \sim p(\cdot|s_t, a_t)$.

We operate under a sparse reward setting, where the lower-level policy is driven by a sparse reward signal $r_t^L(s_t, g_t, a_t) = \mathbf{1}_{\{|s_t - g_t|^2 < \varepsilon\}}$, with $\mathbf{1}_C$ as an indicator function returning 1 if the condition $C$ holds, indicating that the subgoal is reached. The reward function $r^L : \mathcal{S} \times \mathcal{G} \times \mathcal{A} \to \mathbb{R}$ governs the lower level. In contrast, the higher-level policy is rewarded based on $r_t^H(s_t, g^*, g_t)$, where $g^* \in \mathcal{G}$ is the final goal. The reward function for the higher level, $r^H : \mathcal{S} \times \mathcal{G} \times \mathcal{G} \to \mathbb{R}$, encourages progress toward the final goal. Experience for the lower-level policy is stored in its replay buffer

as tuples $(s_t, g_t, a_t, r_t^L, s_{t+1})$, while the higher-level policy stores transitions as $(s_t, g^*, g_t, r_i^H = \sum_{i=t}^{t+k-1} r_i^L, s_{t+k})$, updated every $k$ timesteps. Notably, rewards for the higher-level policy $r_t^H(\pi_L)$ depend on the lower-level policy $\pi_L$. Next, we enlist the challenges of standard HRL methods.

**Challenges of HRL** While HRL offers advantages over traditional RL, such as better sample efficiency through temporal abstraction and enhanced exploration (Nachum et al., 2018; 2019), it also faces significant challenges. This paper focuses on two key issues:

**C1**. Training instability due to *non-stationarity* in off-policy HRL, and
**C2**. Suboptimal performance due to the generation of *infeasible subgoals* by the higher-level policy.

Off-policy HRL suffers from non-stationarity because the behavior of the lower-level policy changes over time (Nachum et al., 2018; Levy et al., 2018). This makes previously collected higher-level transitions in the replay buffer outdated, reducing their effectiveness. Furthermore, the higher-level policy may suggest infeasible subgoals for the lower-level policy (Chane-Sane et al., 2021), hindering learning and decreasing overall performance. Thus, despite its theoretical advantages, HRL often underperforms in practice (Nachum et al., 2018; Levy et al., 2018).

Preference-based learning (PbL) methods such as RLHF (Christiano et al., 2017; Ibarz et al., 2018; Lee et al., 2021) and DPO (Rafailov et al., 2024b) leverage preference data to solve complex tasks. Prior work (Singh et al., 2024) leverages RLHF to mitigate non-stationarity in HRL using preferences. However, directly applying PbL techniques to HRL remains challenging. We now first discuss PbL approaches and then discuss their key limitations when directly applying to HRL.

### 3.1 PbL: Preference Based Learning

**Reinforcement Learning from Human Feedback (RLHF):** In this setting, the agent behavior is represented as a $k$-length trajectory denoted as $\tau$, of state observations and actions: $\tau = ((s_t, a_t), (s_{t+1}, a_{t+1})...(s_{t+k-1}, a_{t+k-1}))$. The reward model to be learned is denoted by $\widehat{r}_\phi : \mathcal{S} \times \mathcal{A} \to \mathbb{R}$, where $\phi$ are the reward model parameters. Thus, the preferences between two trajectories, $\tau^1$ and $\tau^2$, can be expressed through the Bradley-Terry model (Bradley and Terry, 1952):

$$P_\phi \left[ \tau^1 \succ \tau^2 \right] = \frac{\exp \sum_t \widehat{r}_\phi \left( s_t^1, a_t^1 \right)}{\sum_{i \in \{1,2\}} \exp \sum_t \widehat{r}_\phi \left( s_t^i, a_t^i \right)}, \tag{1}$$

where $\tau^1 \succ \tau^2$ implies that $\tau^1$ is preferred over $\tau^2$. The preference dataset $\mathcal{D}$ has entries of the form $(\tau^1, \tau^2, y)$, where $y = (1, 0)$ when $\tau^1$ is preferred over $\tau^2$, $y = (0, 1)$ when $\tau^2$ is preferred over $\tau^1$, and $y = (0.5, 0.5)$ in case of no preference. In RLHF, we first learn the reward function $\widehat{r}_\phi$ (Christiano et al., 2017) using cross-entropy loss along with use equation 1 to yield the formulation:

$$\mathcal{L}(\phi) = -\mathbb{E}_{(\tau^1, \tau^2, y) \sim \mathbb{D}} \left[ \log \sigma(\sum_{t=0}^{T-1} \widehat{r}_\phi \left( s_t^1, g^*, g_t^1 \right) - \sum_{t=0}^{T-1} \widehat{r}_\phi \left( s_t^2, g^*, g_t^2 \right)) \right]. \tag{2}$$

Subsequently, the reward model is used to learn the corresponding policy using RL.

**Direct Preference Optimization (DPO):** Although RLHF provides an elegant framework for learning policies from preferences, it involves RL training step which is often expensive and unstable in practice. In contrast, DPO (Rafailov et al., 2024b) circumvents the need for RL step by using a closed-form solution for the optimal policy of the KL-regularized RL problem (Levine, 2018): $\pi^*(a|s) = \frac{1}{Z(s)} \pi_{ref}(a|s) e^{r(s,a)}$, where $\pi_{ref}$ is the reference policy, $\pi^*$ is the optimal policy, and $Z(s)$ is a normalizing partition function ensuring that $\pi^*$ provides a valid probability distribution over $\mathcal{A}$ for each $s \in \mathcal{S}$. This formulation is rearranged to yield the reward function: $r(s, a) = \alpha \log \pi^*(a|s) - \alpha \log \pi_{ref}(a|s) - Z(s)$, which is then substituted in the standard cross-entropy loss equation 2, to yield the following objective:

$$\mathcal{L}_{DPO}(\theta) = -\mathbb{E}_{(s, y_1, y_2) \sim \mathbb{D}} \left[ \log \sigma \left( \alpha \log \frac{\pi_\theta(y_1|s)}{\pi_{ref}(y_1|s)} - \alpha \log \frac{\pi_\theta(y_2|s)}{\pi_{ref}(y_2|s)} \right) \right], \tag{3}$$

where $\theta$ are the policy parameters and $\sigma$ denotes the sigmoid function. Next we discuss key limitations when directly applying PbL to HRL.

### 3.1.1 LIMITATIONS OF DIRECTLY APPLYING PBL TO HRL

**Directly using `RLHF`**: Prior approaches leverage the advancements in PbL to mitigate non-stationarity (Singh et al., 2024) by utilizing the reward model $r_\phi^H$ learned using `RLHF` as higher level rewards instead of environment rewards $r_{\pi_L}^H$ used in vanilla `HRL` approaches, which depend on the sub-optimal lower primitive. However, such approaches may lead to degenerate solutions by generating infeasible subgoals for the lower-level primitive. Additionally, such approaches require `RL` as an intermediate step, which might cause training instability (Rafailov et al., 2024b).

**Directly using `DPO`**: In temporally extended task environments like robotics, directly extending DPO to the `HRL` framework is also non-trivial due to three reasons: $(i)$ such scenarios deal with multi-step trajectories involving stochastic transitions models, $(ii)$ efficient pre-trained reference policies are typically unavailable in robotics, $(iii)$ similar to `RLHF`, such approaches may produce degenerate solutions when the higher level policy subgoal predictions are infeasible.

## 4 HPO : HIERARCHICAL PREFERENCE OPTIMIZATION

To deal with the challenges of `HRL` and utilizing `PbL` in `HRL` framework (3.1.1), we introduce `HPO` : **H**ierarchical **P**reference **O**ptimization, a novel approach that leverages primitive regularized `DPO` to mitigate non-stationarity and infeasible subgoal generation issues in HRL. The rest of this section proceeds as follows: we first critically analyze the key source of non-stationarity in `HRL` and formulate HRL as a bi-level optimization problem. Utilizing this formulation, we then derive a primitive-regularized token-level `DPO` objective that overcomes the limitations of non-stationarity and infeasible subgoals generation in HRL. Finally, we analyze the `HPO` objective and gradient, and provide the final algorithm.

### 4.1 HRL: BI-LEVEL FORMULATION

We first consider the bi-level formulation for `HRL`. Let $T$ be the task horizon, $s_t$ be the state at time $t$, $g^*$ be the final goal, and $g_t \sim \pi^H(.|s_t, g^*)$ be the higher level subgoal prediction at time $t$. In the `HRL` framework, the lower level optimal policy is represented as $\pi_*^L$. The higher-level policy $\pi^H$ predicts subgoal $g_t$ to $\pi_*^L$, and receives higher level rewards $r_{\pi_*^L}^H$ (which depends on the optimal lower primitive $\pi_*^L$). The lower primitive policy executes primitive actions $a_t$ to achieve the predicted subgoals. The higher level objective can thus be represented as:

$$\max_{\pi^H} \mathbb{E}_{\pi^H}\left[ \sum_{t=0}^{T-1} r_{\pi_*^L(\cdot|s_t,g_t)}^H (s_t, g^*, g_t) \right]. \tag{4}$$

**Issue of non-stationarity.** In vanilla `HRL`, when the hierarchical levels are trained concurrently, since we do not have access to the optimal $\pi_*^L(\cdot|s, g)$ policy in the objective in equation 4, the higher level rewards are instead generated using an approximated lower primitive $\pi_\theta^L(\cdot|s, g)$, which is sub-optimal. Due to this sub-optimal lower level primitive, the higher level rewards $r_{\pi_\theta^L}^H$ are non-stationary (e.g. we may encounter different higher level rewards for the same subgoal prediction, as $\pi_\theta^L(\cdot|s, g)$ changes). This causes the non-stationarity issue in HRL.

**Issue of infeasible subgoal generation.** Since the sub-optimality in the lower-level policy affects its ability to reach a given subgoal, it consequently impacts the higher-level credit assignment during subsequent subgoal generation. This may cause the higher level to produce infeasible subgoals.

**Distribution shift issue and bi-level formulation.** To understand further the technical reasoning of the above-mentioned issues, let us expand upon the nested structure of the objective in equation 4, which is a bi-level optimization problem given by

$$\max_{\pi^H} \mathcal{J}(\pi^H, \pi_*^L(\pi^H)) \quad s.t. \quad \pi_*^L(\cdot|s_t, g_t) = \arg\max_{\pi^L(\cdot|s_t,g_t)} V_{\pi^L(\cdot|s_t,g_t)}(\pi^H), \forall t \in [0:T-1], \tag{5}$$

where $\mathcal{J}(\pi^H, \pi_*^L(\pi^H))$ represents the higher level maximization objective (cf. equation 4), and $V_{\pi^L}(\pi^H)$ is the lower level value function, conditioned on the higher level policy subgoals. We note that the problem in equation 5 is just an expanded version of the problem in equation 4. From the

bi-level structure in equation 5, we note that when we approximate the optimal lower level policy $\pi_L^*$ with an approximated version $\pi_\theta^L$, it would actually break the nested structure of the problem in equation 5, leading to the issues of non-stationarity as well as infeasible subgoal generation. In this work, we propose a novel approach to solving the original HRL problem in equation 5 without breaking the nested structure as follows.

**Directly considering the nested structure in equation 4.** Utilizing the recent advancements in the optimization literature (Liu et al., 2022), we consider the equivalent constraint formulation of the problem in equation 5 as follows

$$\max_{\pi^H, \pi^L} \mathcal{J}(\pi^H, \pi^L) \quad s.t. \quad V_{\pi^L(\cdot|s_t, g_t)}(\pi^H) \geq V_{\pi_*^L(\cdot|s_t, g_t)}(\pi^H), \forall t \in [0 : T-1]. \tag{6}$$

The constraint optimization problem is also challenging because it requires the access to $V_{\pi_*^L}(\pi^H)$ which is hard to obtain in practice. But let us try to discuss the constraint in detail. The constraint $V_{\pi^L}(\pi^H) \geq V_{\pi_*^L}(\pi^H)$ is essentially to enforce the optimality of the lower level policy, because $V_{\pi_*^L}(\pi^H)$ is the maximum (optimal) value. But we note that for the sparse reward scenarios (wlog assuming reward is between 0 and 1), the value function $V_{\pi^L}(\pi^H) = 0$ for any non-goal reaching policy. But $V_{\pi_*^L}(\pi^H) > 0$ or equivalently is definitely non-zero for the goal reaching optimal policy $\pi_*^L$. Therefore, it is sufficient to ensure $V_{\pi^L}(\pi^H) > 0$ in the constraint. To make the problem tractable, we introduce a parameter delta and reformulate the problem in equation 6 as follows:

$$\max_{\pi^H, \pi^L} \mathcal{J}(\pi^H, \pi^L) \quad s.t. \quad V_{\pi^L(\cdot|s_t, g_t)}(\pi^H) \geq \delta, \forall t \in [0 : T-1], \tag{7}$$

where $\delta > 0$. Now, we formulate equation 7 as the following Lagrangian objective and substitute the objective from equation 4, where $\lambda$ is the regularization weight hyper-parameter:

$$\max_{\pi^H, \pi^L} \mathbb{E}_{\pi^H} \left[ \sum_{t=0}^{T-1} r_{\pi^L(\cdot|s_t, g_t)}^H(s_t, g^*, g_t) + \lambda_t (V_{\pi^L(\cdot|s_t, g_t)}(\pi^H) - \delta) \right]. \tag{8}$$

We can use equation 8 to solve the HRL policies for both higher and lower level, where $(i)$ the higher level policy learns to achieve the final goal and predict feasible subgoals to the lower level policy, and $(ii)$ the lower level policy learns to achieve the predicted subgoals. This will mitigate the issues of non-stationarity (**C1**) and infeasible subgoal generation (**C2**). However, directly optimizing equation 8 requires knowledge of higher level reward function $r_{\pi^L}^H$ as a function of $\pi^L$, which is unavailable. Also as mentioned before, using RL to optimize the objective will be unstable.

Next, we discuss `HPO`: our `PbL` based approach that efficiently optimizes equation 8 by learning a primitive-regularized `DPO` objective which simultaneously solves the issues of non-stationarity (**C1**) and infeasible subgoal generation (**C2**).

## 4.2 HPO

Here, we derive our `HPO` objective. When optimizing for the higher level policy, equation 8 can be re-written as:

$$\max_{\pi^H} \mathbb{E}_{\pi^H} \left[ \sum_{t=0}^{T-1} \widehat{r}_\phi \right], \tag{9}$$

where, $\widehat{r}_\phi = r_L^H(s_t, g^*, g_t) + \lambda(V_{\pi^L}(s_t, g_t) - \delta)$ (for simplicity, we will henceforth represent $r_{\pi^L(s_t, g_t)}^H$ by $r_L^H$). As originally explored in Garg et al. (2021), the relationship between the future returns and the current timestep return can be captured by the following bellman equation corresponding to the optimal higher level policy $\pi_*^H$:

$$Q_*^H(s_t, g^*, g_t) = \begin{cases} r_L^H(s_t, g^*, g_t) + V_*^H(s_{t+1}, g^*) & \text{if } s_{t+1} \text{ is not terminal,} \\ r_L^H(s_t, g^*, g_t) & \text{if } s_{t+1} \text{ is terminal.} \end{cases} \tag{10}$$

Since there is a bijection between the reward function $r_L^H(s_t, g^*, g_t)$ and corresponding optimal state value function $Q_*^H(s_t, g^*, g_t)$ in the token MDP (Rafailov et al., 2024a), we reformulate equation 10 to represent the rewards as follows:

$$r_L^H(s_t, g^*, g_t) = \left[ Q_*^H(s_t, g^*, g_t) - V_*^H(s_{t+1}, g^*) \right]. \tag{11}$$

For the full trajectory $\tau$, we can use equation 9 and equation 11 to get:

$$\sum_{t=0}^{T-1} \widehat{r}_\phi = \sum_{t=0}^{T-1} \left( Q_*^H(s_t, g^*, g_t) - V_*^H(s_{t+1}, g^*) + \lambda(V_{\pi^L}(s_t, g_t) - \delta) \right)$$

$$\stackrel{(a)}{=} V_*^H(s_0, g^*) + \sum_{t=0}^{T-1} \left( Q_*^H(s_t, g^*, g_t) - V_*^H(s_t, g^*) + \lambda(V_{\pi^L}(s_t, g_t) - \delta) \right)$$

$$\stackrel{(b)}{=} V_*^H(s_0, g^*) + \sum_{t=0}^{T-1} \left( A_*^H(s_t, g^*, g_t) + \lambda(V_{\pi^L}(s_t, g_t) - \delta) \right)$$

$$\stackrel{(c)}{=} V_*^H(s_0, g^*) + \sum_{t=0}^{T-1} \left( \beta \log \pi_*^H(g_t|s_t, g^*) + \lambda(V_{\pi^L}(s_t, g_t) - \delta) \right), \tag{12}$$

where (a) is due to adding and subtracting $V_*^H(s_0, g^*)$, (b) is due to $Q_*^H(s_t, g^*, g_t) - V_*^H(s_t, g^*) = A_*^H(s_t, g^*, g_t)$ where $A_*^H(s_t, g^*, g_t)$ represents the advantage function, and (c) is due to a result from (Ziebart, 2010), $A_*^H(s_t, g^*, g_t) = \beta \log(\pi_*^H(g_t|s_t, g^*))$. Substituting equation 12 in equation 2 yields the following primitive regularized `DPO` objective:

$$\mathcal{L}(\pi_*^H, \mathbb{D}) = -\mathbb{E}_{(\tau^1, \tau^2, y) \sim \mathbb{D}} \left[ \log \sigma \left( \sum_{t=0}^{T-1} \left( \beta \log \pi_*^H(g_t^1|s_t^1, g^*) - \beta \log \pi_*^H(g_t^2|s_t^2, g^*) \right. \right. \right.$$

$$\left. \left. \left. + \lambda V_{\pi^L}(s_t^1, g_t^1) - \lambda V_{\pi^L}(s_t^2, g_t^2) \right) \right) \right]. \tag{13}$$

Note that terms $V_*^H(s_0, g^*)$ and $\lambda\delta$ are same for both trajectories and hence they cancel. This is the final `HPO` objective which optimizes the higher-level policy using primitive regularized `DPO`. Next, we analyze and discuss the `HPO` gradient.

**Analyzing `HPO` gradient:** We further analyze the `HPO` objective by computing and interpreting the gradients of $\mathcal{L}(\pi_*^H, \mathbb{D})$ with respect to higher level policy $\pi_*^H$:

$$\nabla \mathcal{L}(\pi_*^H, \mathbb{D}) = -\beta \mathbb{E}_{(\tau_1, \tau_2, y) \sim \mathbb{D}} \left[ \sum_{t=0}^{T-1} \left( \underbrace{\sigma\left(\hat{r}\left(s_t^2, g_t^2\right) - \hat{r}\left(s_t^1, g_t^1\right)\right)}_{\text{higher weight for incorrect preference}} \right. \right.$$

$$\left. \left. \cdot \left( \underbrace{\nabla \log \pi^H\left(g_t^1|s_t^1, g^*\right)}_{\text{increase likelihood of } \tau_1} - \underbrace{\nabla \log \pi^H\left(g_t^2|s_t^2, g^*\right)}_{\text{decrease likelihood of } \tau_2} \right) \right) \right]. \tag{14}$$

where $\hat{r}(s_t, g_t, g^*) = \beta \log \pi^H(g_t|s_t, g^*) + \lambda V_{\pi^L}(s_t, g_t)$, which is the implicit reward determined by the higher-level policy and the lower-level value function. Conceptually, this objective increases the likelihood of preferred trajectories while decreasing the likelihood of dispreferred trajectories. Further, according to the strength of the KL constraint, the examples are weighted based on how inaccurately the implicit reward model $\hat{r}(s_t, g_t, g^*)$ ranks the trajectories. Notably, the $\lambda$ weighted value function term $V_{\pi^L}$ in the implicit reward $\hat{r}$ assigns high value to feasible subgoals, thus enabling primitive regularization.

**Replacing human feedback:** In the vanilla `PbL` framework, preferences are elicited from human feedback (Christiano et al., 2017). However, collecting large amount of human preference feedback

required for `PbL` is computationally expensive. In this work, we follow the primitive-in-the-loop (PiL) approach in PIPER (Singh et al., 2024), and simulate this preference feedback by using implicit sparse rewards: $r^s(s_t, g^*, g_t)$, to determine preferences $y$ between trajectories $\tau^1$ and $\tau^2$. This feedback is obtained as follows: suppose the higher level policy $\pi^H$ predicts subgoal $g_t \sim \pi^H(\cdot | s_t, g^*)$ for state $s_t$ and final goal $g^*$. The lower-level primitive executes primitive actions according to its policy $\pi^L$ for $k$ timesteps to end up in state $s_{t+k-1}$. We use the sparse reward provided by the environment at state $s_{t+k-1}$ as the implicit sparse reward, i.e., $r^s(s_t, g^*, g_t) = \mathbf{1}_{\{\|s_{t+k-1}-g^*\|_2 \leq \epsilon\}}$. We use the sum of all k-step rewards $r^s$ for comparing the two trajectories, to obtain preferences between higher level behavior sequences, as explained in detail in Singh et al. (2024).

We provide the psuedo-code for `HPO` algorithm in the Appendix A.1 Algorithm 1.

## 5 EXPERIMENTS

In our empirical analysis, we ask the following questions:
**(1) How well does `HPO` perform against baselines?**: How well does `HPO` perform in complex robotics control tasks against prior hierarchical and non-hierarchical baselines?
**(2) Does `HPO` mitigate HRL limitations?**: How well does `HPO` mitigate the issues of non-stationarity and infeasible subgoal generation in `HRL`?
**(3) What is the impact of our design decisions on the overall performance?**: Can we concretely justify our design choices through extensive ablation analysis?

**Task details**: Here, we explain the experimental setup and test beds for comparing `HPO` against baselines. We assess `HPO` on four robotic navigation and manipulation environments: ($i$) maze navigation, ($ii$) pick and place (Andrychowicz et al., 2017), ($iii$) push, and ($iv$) franka kitchen environment (Gupta et al., 2019). These are formulated as sparse reward scenarios, where the agent is only rewarded when it comes within a $\delta$ distance of the goal. Due to this, these environments are hard where the agent must extensively explore the environment before coming across any rewards. As an example: in franka kitchen task, the agent only receives a sparse reward after achieving the final goal (e.g. successfully opening the microwave and then turning on the gas knob).

**Environment details**: Since in our goal-conditioned `RL` setting, the final goals are randomly generated, and this further increases the task complexity. As empirically demonstrated later, these challenges prevent previous baselines from performing well, making these environments ideal test beds for evaluating the advantages of efficient hierarchical preference-based hierarchical policy learning. Unless otherwise stated, we maintain consistency across all baselines to ensure fair comparisons. Finally, for the more complex sparse reward tasks such as pick and place, push, and franka kitchen, we assume access to one human demonstration and incorporate an imitation learning objective at the lower level to accelerate learning. However, we do not use demonstrations in the maze task. However, we apply the same assumption consistently across all baselines to ensure fairness. Further implementation and environment details are provided in Appendix A.3 and A.5, and the implementation code is provided in the supplementary.

### 5.1 HOW WELL DOES `HPO` PERFORM AGAINST BASELINES?

In this section, we compare `HPO` against multiple hierarchical and non-hierarchical baselines. Please refer to Figure 2 for success rate comparison plots and subsequent discussion. The solid line and shaded regions represent the mean and standard deviation, averaged over 5 seeds.

**`HPO-No-V` baseline:** `HPO-No-V` is an ablation of `HPO` without primitive regularization, which uses `DPO` at the higher level to predict subgoals, and `RL` at the lower level policy. Since primitive regularization conditions the higher level policy into predicting feasible subgoals, we use this baseline to demonstrate the importance of feasible subgoal generation. As seen in Figure 2, although this baseline mitigates non-stationarity using preference based learning and outperforms prior baselines, `HPO` outperforms this baseline demonstrating that both non-stationarity mitigation and feasible subgoal generation are crucial for improved performance.

**`HPO-FLAT` baseline:** Here, we compare `HPO` with `HPO-FLAT`, which is a token-level `DPO` (Rafailov et al., 2024a) implementation for our sparse reward robotics tasks. Note that since we do not have access to a pre-trained model as a reference policy in robotics scenarios like generative

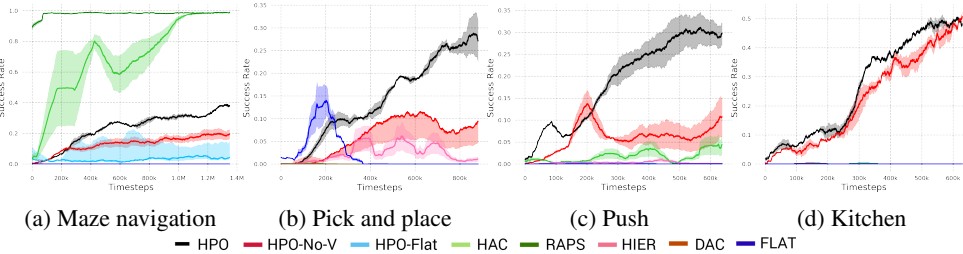

(a) Maze navigation    (b) Pick and place    (c) Push    (d) Kitchen

— HPO — HPO-No-V — HPO-Flat — HAC — RAPS — HIER — DAC — FLAT

Figure 2: **Success Rate plots** This figure illustrates the success rates across four sparse-reward maze navigation and robotic manipulation tasks, where the solid lines represent the mean, and the shaded areas denote the standard deviation across 5 different seeds. We evaluate `HPO` against several baselines. Although `HAC` and `RAPS` surpass `HPO` in the maze task, `HPO` demonstrates strong performance and significantly outperforms the baselines in the more challenging tasks.

language modeling, we use a uniform policy as a reference policy, which effectively translates to an additional objective of maximizing the entropy of the learnt policy. `HPO` is an hierarchical approach which benefits from temporal abstraction and improved exploration, as is apparent from Figure 2 which shows that `HPO` significantly outperforms this baseline.

**RAPS baseline:** Here, we consider `RAPS` (Dalal et al., 2021) baseline, which employs behavior priors at the lower level, effectively simplifying the hierarchical structure for solving the task. Although `RAPS` is an elegant framework for solving robotic tasks where behavior priors are readily available, it requires considerable effort to construct such priors and struggles to perform well in their absence, especially when dealing with sparse reward scenarios. Indeed we empirically find this to be the case, since although `RAPS` performs exceptionally well in maze navigation task, it fails to perform well in other sparse complex manipulation tasks.

**HAC baseline:** In order to analyze how `HPO` compares with prior approaches that tackle non-stationarity, we also consider `HAC` (Levy et al., 2018) baseline. `HAC` tries to mitigate non-stationarity by simulating optimal lower level primitive behavior. This approach performs relabeling on the replay buffer transitions and is closely related to hindsight experience replay (`HER`) (Andrychowicz et al., 2017). Although `HAC` performs well in maze navigation task, it struggles to perform well in harder manipulation environments. `HPO` outperforms this baseline in 3 of 4 tasks.

**HIER baseline:** We also implement `HIER`, a vanilla `HRL` baseline implemented using `SAC` (Haarnoja et al., 2018) at both hierarchical levels to demonstrate the significance of dealing with non-stationarity. As expected, `HPO` is able to consistently outperform this baseline in all tasks.

**DAC baseline:** We implement a single-level baseline: Discriminator Actor Critic (`DAC`) (Kostrikov et al., 2018). We provide one demonstration to `DAC` baseline in each environment. However, despite having access to privileged information, `DAC` still struggles to perform well.

**FLAT baseline:** We also implement a `FLAT` baseline using single-level `SAC`. As seen in Figure 2, this baseline also fails to achieve good results, highlighting the importance of our hierarchical structure for success in complex robotic tasks.

## 5.2 DOES HPO MITIGATE HRL LIMITATIONS?

In this section, we empirically verify whether `HPO` indeed mitigates the non-stationarity and infeasible subgoal generation limitations of `HRL`.

**C1: Non-stationarity issue in HRL** Here, we analyze whether `HPO` mitigates non-stationarity in `HRL`, by comparing against vanilla `HIER` baseline in Figure 3. We compute average distance between subgoals predicted by the higher level policy and subgoals achieved by the lower level primitive for 100 rollouts during various stages of training. If `HPO` indeed mitigates non-stationarity, this average distance should be low, since this implies that the `HPO` predicts subgoals achievable by the lower primitive and hence induces optimal lower primitive behavior. Indeed, we find that `HPO` consistently generates low average distance values, which implies that `HPO` mitigates non-stationarity.

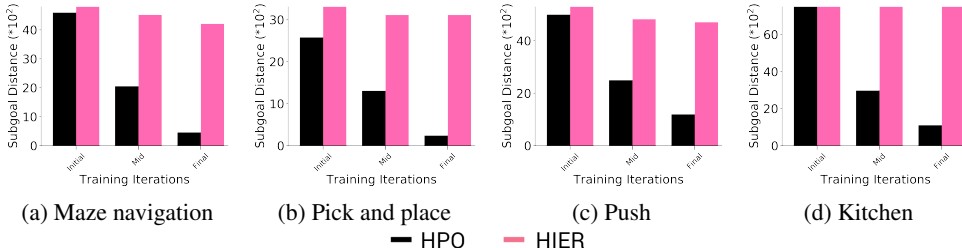

(a) Maze navigation  (b) Pick and place  (c) Push  (d) Kitchen

Figure 3: **Non-stationarity ablation** This figure compares HPO with HIER baseline, based on average distance between subgoals predicted by the higher level policy and subgoals achieved by the lower level primitive *during training*. HPO consistently generates low average distance values, which implies that in HPO, the higher level policy generates achievable subgoals that induce optimal lower primitive goal reaching behavior. This mitigates non-stationary in HRL and leads to improved performance.

**C2: Infeasible subgoal generation issue in HRL** As seen in Figure 4, after training, the average distance values for HPO are significantly lower than HIER baseline, which implies that HPO generates feasible subgoals that are achievable by the lower primitive due to primitive regularization.

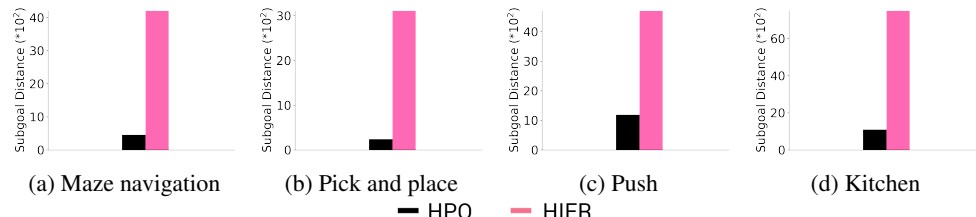

(a) Maze navigation  (b) Pick and place  (c) Push  (d) Kitchen

Figure 4: **Feasible subgoal generation ablation** This figure compares HPO with HIER baseline, based on the average distance between the subgoals predicted by the higher level policy and the subgoals achieved by the lower level policy *after the training is completed*. As seen in figure, the average distance values for HPO are significantly lower than HIER baseline, which implies that HPO generates feasible subgoals that are achievable by the lower primitive.

### 5.3 WHAT IS THE IMPACT OF OUR DESIGN DECISIONS ON THE OVERALL PERFORMANCE?

In this section, we conduct experiments to evaluate the impact of each individual design choice. Concretely, we provide ablation studies and insights on selecting the max-ent parameter $\beta$ and the regularization weight $\lambda$, as shown in Appendix A.2 Figures 5 and 6. Please refer to the Appendix for more detailed explanation.

## 6 CONCLUSION

In this work, we introduce Hierarchical Preference Optimization HPO, a novel hierarchical approach that employs primitive-regularized direct preference optimization DPO to mitigate the issues of non-stationarity and infeasible generation in HRL. HPO employs token-level DPO to efficiently learn higher level policy using preference data, and RL to learn the lower level primitive policy. Since the DPO formulation avoids the dependence on changing lower level primitive, it mitigates non-stationarity. We formulate HRL as a bi-level optimization objective to insure that the higher level policy generates feasible subgoals, and avoids degenerate solutions. Based on strong empirical results, We believe that HPO is an important step towards learning effective control policies for solving complex sparse robotics scenarios.

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

CONTENTS

## A APPENDIX

### A.1 HPO ALGORITHM

Here, we provide the pseudo-code for HPO algorithm:

---
**Algorithm 1** HPO
---
1: Initialize preference dataset $\mathcal{D} = \{\}$.
2: Initialize lower level replay buffer $\mathcal{R}^L = \{\}$.
3: **for** $i = 1 \ldots N$ **do**
4:     // Collect higher level trajectories $\tau$ using $\pi^H$ and lower level trajectories $\rho$ using $\pi^L$,
5:     // and store the trajectories in $\mathcal{D}$ and $\mathcal{R}^L$ respectively.
6:     // After every m timesteps, relabel $\mathcal{D}$ using primitive-in-the-loop feedback $y$.
7:     // Lower level value function update
8:     Optimize lower level value function $V_{\pi_L}$ to get $V_{\pi_L}^k$.
9:     // Higher level policy update using HPO
10:     **for** each gradient step **do**
11:         // Sample higher level behavior trajectories.
12:         $(\tau^1, \tau^2, y) \sim \mathcal{D}$
13:         Optimize higher level policy $\pi^U$ using equation 13.
14:     // Lower primitive policy update using `RL`
15:     **for** each gradient step **do**
16:         Sample $\rho$ from $\mathcal{R}^L$.
17:         Optimize lower policy $\pi^L$ using SAC.

---

### A.2 ABLATION EXPERIMENTS

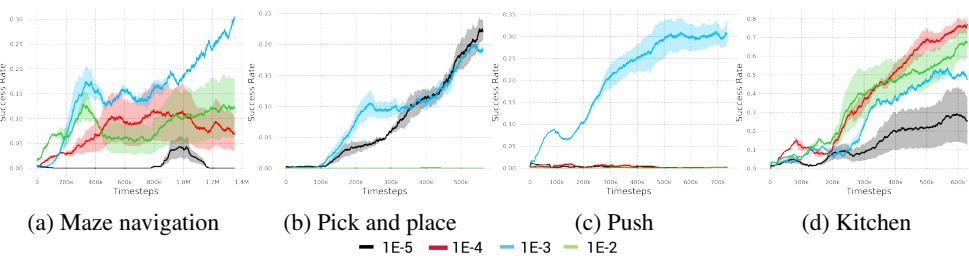

   (a) Maze navigation       (b) Pick and place       (c) Push       (d) Kitchen

           1E-5    1E-4    1E-3    1E-2

Figure 5: **Regularization weight ablation** This figure depicts the success rate performance for varying values of the primitive regularization weight $\lambda$. When $\alpha$ is too small, we loose the benefits of primitive-informed regularization resulting in poor performance, whereas too large $\alpha$ values can lead to degenerate solutions. Hence, selecting appropriate $\lambda$ value is essential for accurate subgoal prediction and enhancing overall performance.

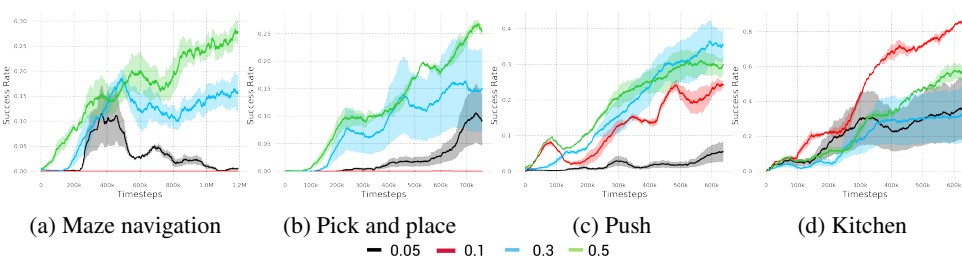

   (a) Maze navigation       (b) Pick and place       (c) Push       (d) Kitchen

           0.05    0.1    0.3    0.5

Figure 6: **Max-ent parameter ablation** This figure illustrates the success rate performance for different values of the max-ent parameter $\beta$ hyper-parameter. This parameter controls the exploration in maximum-entropy formulation. If $\beta$ is too large, the higher-level policy may perform extensive exploration but stay away from optimal subgoal prediction, whereas if $\beta$ is too small, the higher-level might not explore and predict sub-optimal subgoals. Hence, selecting appropriate $\beta$ value is essential for enhancing overall performance.

Here, we perform ablations to analyze our design choices. We first analyze the effect of varying regularization weight $\lambda$ hyper-parameter in Figure 5. $\lambda$ controls the strength of primitive regularization: if $\lambda$ is too small, we lose the benefit of primitive regularization and the higher level policy may predict infeasible subgoals, whereas if $\lambda$ is too large, the higher level policy may fail to achieve the final goal and keep predicting simple feasible subgoals. Therefore, selecting a good $\lambda$ value is crucial. Next, we analyze the effect of varying $\beta$ hyper-parameter in Figure 5. If $\beta$ is too large, the policy keeps exploring without predicting optimal subgoal, whereas if $\beta$ is too small, the policy might not explore which may lead to sub-optimal subgoal predictions. Therefore, we perform this ablation to select good $\lambda$ value in all the environments.

### A.3    IMPLEMENTATION DETAILS

We conducted experiments on two systems, each equipped with Intel Core i7 processors, 48GB of RAM, and Nvidia GeForce GTX 1080 GPUs. The experiments included the corresponding timesteps taken for each run. For the environments $(i) - (iv)$, the maximum task horizon $\mathcal{T}$ is set to 225, 50, 50, and 225 timesteps, respectively, with the lower-level primitive allowed to execute for 15, 7, 7, and 15 timesteps. We used off-policy Soft Actor Critic (SAC)(Haarnoja et al., 2018) to optimize the RL objective, leveraging the Adam optimizer(Kingma and Ba, 2014). Both the actor and critic networks consist of three fully connected layers with 512 neurons per layer. The total timesteps for experiments in environments $(i) - (iv)$ are 1.35e6, 9e5, 1.3E6, and 6.3e5, respectively.

For the maze navigation task, a 7-degree-of-freedom (7-DoF) robotic arm navigates a four-room maze with its gripper fixed at table height and closed, maneuvering to reach a goal position. In the pick-and-place task, the 7-DoF robotic arm gripper locates, picks up, and delivers a square block to the target location. In the push task, the arm's gripper must push the square block toward the goal. For the kitchen task, a 9-DoF Franka robot is tasked with opening a microwave door as part of a predefined complex sequence to reach the final goal. We compare our approach with the Discriminator Actor-Critic (Kostrikov et al., 2018), which uses a single expert demonstration. Although this study doesn't explore it, combining preference-based learning with demonstrations presents an exciting direction for future research (Cao et al., 2020).

To ensure fair comparisons, we maintain uniformity across all baselines by keeping parameters such as neural network layer width, number of layers, choice of optimizer, SAC implementation settings, and others consistent wherever applicable. In RAPS, the lower-level behaviors are structured as follows: for maze navigation, we design a single primitive, *reach*, where the lower-level primitive moves directly toward the subgoal predicted by the higher level. For the pick-and-place and push tasks, we develop three primitives: *gripper-reach*, where the gripper moves to a designated position $(x_i, y_i, z_i)$; *gripper-open*, which opens the gripper; and *gripper-close*, which closes the gripper. In the kitchen task, we use the action primitives implemented in RAPS (Dalal et al., 2021).

#### A.3.1    ADDITIONAL HYPER-PARAMETERS

Here, we enlist the additional hyper-parameters used in HPO:

```
activation:  tanh [activation for reward model]
layers:  3 [number of layers in the critic/actor networks]
hidden:  512 [number of neurons in each hidden layers]
Q_lr:  0.001 [critic learning rate]
pi_lr:  0.001 [actor learning rate]
buffer_size:  int(1E7) [for experience replay]
clip_obs:  200 [clip observation]
n_cycles:  1 [per epoch]
n_batches:  10 [training batches per cycle]
batch_size:  1024 [batch size hyper-parameter]
reward_batch_size:  50 [reward batch size for DPO-FLAT]
random_eps:  0.2 [percentage of time a random action is taken]
alpha:  0.05 [weightage parameter for SAC]
noise_eps:  0.05 [std of gaussian noise added to
not-completely-random actions]
norm_eps:  0.01 [epsilon used for observation normalization]
norm_clip:  5 [normalized observations are cropped to this values]
```

```
adam_beta1:  0.9 [beta 1 for Adam optimizer]
adam_beta2:  0.999 [beta 2 for Adam optimizer]
```

### A.4 IMPACT STATEMENT

Our proposed approach and algorithm are not expected to lead to immediate technological advancements. Instead, our primary contributions are conceptual, focusing on fundamental aspects of Hierarchical Reinforcement Learning (HRL). By introducing a preference-based methodology, we offer a novel framework that we believe has significant potential to enhance HRL research and its related fields. This conceptual foundation paves the way for future investigations and could stimulate advancements in HRL and associated areas.

### A.5 ENVIRONMENT DETAILS

#### A.5.1 MAZE NAVIGATION TASK

In this environment, a 7-DOF robotic arm gripper must navigate through randomly generated four-room mazes. The gripper remains closed, and both the walls and gates are randomly placed. The table is divided into a rectangular $W \times H$ grid, with vertical and horizontal wall positions $W_P$ and $H_P$ selected randomly from the ranges $(1, W - 2)$ and $(1, H - 2)$, respectively. In this four-room setup, gate positions are also randomly chosen from $(1, W_P - 1)$, $(W_P + 1, W - 2)$, $(1, H_P - 1)$, and $(H_P + 1, H - 2)$. The gripper's height remains fixed at table height, and it must move through the maze to reach the goal, marked by a red sphere.

For both higher and lower-level policies, unless stated otherwise, the environment consists of continuous state and action spaces. The state is encoded as a vector $[p, \mathcal{M}]$, where $p$ represents the gripper's current position, and $\mathcal{M}$ is the sparse maze representation. The input to the higher-level policy is a concatenated vector $[p, \mathcal{M}, g]$, with $g$ representing the goal position, while the lower-level policy input is $[p, \mathcal{M}, s_g]$, where $s_g$ is the subgoal provided by the higher-level policy. The current position of the gripper is treated as the current achieved goal. The sparse maze array $\mathcal{M}$ is a 2D one-hot vector, where walls are denoted by a value of $1$ and open spaces by $0$.

In our experiments, the sizes of $p$ and $\mathcal{M}$ are set to 3 and 110, respectively. The higher-level policy predicts the subgoal $s_g$, so its action space aligns with the goal space of the lower-level primitive. The lower-level primitive's action, $a$, executed in the environment, is a 4-dimensional vector, where each dimension $a_i \in [0, 1]$. The first three dimensions adjust the gripper's position, while the fourth controls the gripper itself: 0 indicates fully closed, 0.5 means half-closed, and 1 means fully open.

#### A.5.2 PICK AND PLACE AND PUSH ENVIRONMENTS

In the pick-and-place environment, a 7-DOF robotic arm gripper is tasked with picking up a square block and placing it at a designated goal position slightly above the table surface. This complex task involves navigating the gripper to the block, closing it to grasp the block, and then transporting the block to the target goal. In the push environment, the gripper must push a square block towards the goal position. The state is represented by the vector $[p, o, q, e]$, where $p$ is the gripper's current position, $o$ is the block's position on the table, $q$ is the relative position of the block to the gripper, and $e$ contains the linear and angular velocities of both the gripper and the block.

The higher-level policy input is the concatenated vector $[p, o, q, e, g]$, where $g$ denotes the target goal position, while the lower-level policy input is $[p, o, q, e, s_g]$, with $s_g$ being the subgoal provided by the higher-level policy. The current position of the block is treated as the achieved goal. In our experiments, the dimensions for $p$, $o$, $q$, and $e$ are set to 3, 3, 3, and 11, respectively. The higher-level policy predicts the subgoal $s_g$, so the action and goal space dimensions align. The lower-level action $a$ is a 4-dimensional vector, where each dimension $a_i$ falls within the range $[0, 1]$. The first three dimensions adjust the gripper's position, and the fourth controls the gripper itself (0 for closed, 1 for open). During training, the block and goal positions are randomly generated, with the block always starting on the table and the goal placed above the table at a fixed height.

## A.6 ENVIRONMENT VISUALIZATIONS

Here, we provide some visualizations of the agent successfully performing the task.

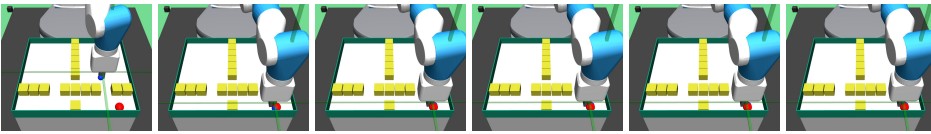

Figure 7: **Maze navigation task visualization**: The visualization is a successful attempt at performing maze navigation task

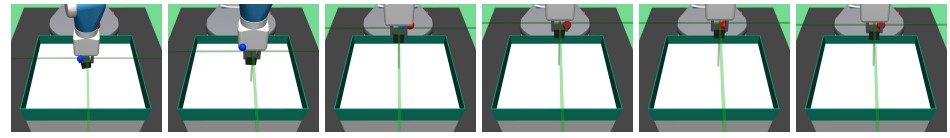

Figure 8: **Pick and place task visualization**: This figure provides visualization of a successful attempt at performing pick and place task

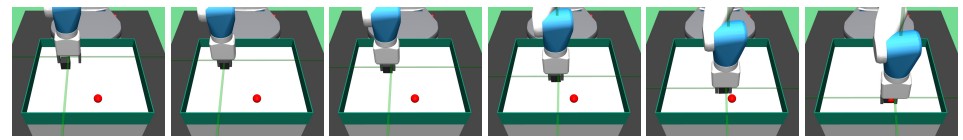

Figure 9: **Push task visualization**: The visualization is a successful attempt at performing push task

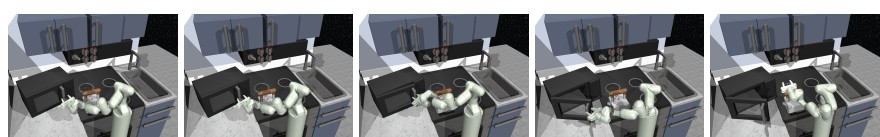

Figure 10: **Kitchen task visualization**: The visualization is a successful attempt at performing kitchen task

