# OpenReview forum: "Hierarchical Preference Optimization: Learning to achieve goals via feasible subgoals prediction"
_ICLR.cc/2025/Conference — ICLR 2025 Conference Withdrawn Submission_

### Official Review · Reviewer_DgAQ · 2024-10-30

**Soundness:** 1
**Presentation:** 3
**Contribution:** 2
**Rating:** 3
**Confidence:** 4

**Summary:**

The paper proposed a Hierarchical Preference Optimization (HPO) algorithm for hierarchical reinforcement learning. The algorithm aims to generate feasible subgoals and mitigate the non-stationary in HRL. HPO leveraged the low-level value functions to condition higher-level policy for subgoal generation and utilized the direct preference optimization (DPO) to optimize the higher-level policy.

**Strengths:**

The idea to introduce low-level value to regularize high-level policy optimization, and leveraging DPO to optimize a traditional RL problem is novel.

**Weaknesses:**

1. The proposed HPO algorithm is built on the goal-conditioned HRL concept. However, in the problem formulation, the definition of the high-level reward deviates from the standard goal-conditioned HRL framework, making it not a robust problem definition. Additionally, some derivations need further clarification or analysis. See details in Questions.

2. The HPO is an HRL approach, but the paper doesn't compare with SOTA HRL works. I encourage the author to involve at least one recently representative HRL algorithm as baseline to further demonstrate HPO's advantages (reference [1][2][3]).

[1] Gürtler, Nico, Dieter Büchler, and Georg Martius. "Hierarchical reinforcement learning with timed subgoals." Advances in Neural Information Processing Systems. (2021).

[2] Kim, Junsu, Younggyo Seo, and Jinwoo Shin. "Landmark-guided subgoal generation in hierarchical reinforcement learning." Advances in neural information processing systems. (2021).

[3] Zhang T, Guo S, Tan T, Hu X, Chen F. Generating adjacency-constrained subgoals in hierarchical reinforcement learning. Advances in Neural Information Processing Systems. (2020).

**Questions:**

**Question 1: Problem Formulation**.

The problem formulation for goal-conditioned HRL is not entirely accurate. Specifically, in the paragraph starting from Line 157: "the lower-level policy is driven by a sparse reward signal, ...., indicating that the subgoal is reached." This is correct, as the low-level policy aims to achieve the subgoal set by the high-level policy. However, the high-level reward function is defined as $r^H = \sum_{sub-trajectory}{r^L}$, where $r^L$ is the low-level reward. This doesn't seem correct to me, as the high-level aims to generate sub-goals guide the low-level to **achieve the final task objective**, i.e., the high-level reward is usually defined based on the environmental reward signal from the problem MDP. (Check the goal-conditioned HRL framework definition in reference [4]). With the definition given in the paper, the high-level reward appears to be evaluating "how many steps in total of the low-level policy is staying near my generated subgoal." In this problem formulation, the original environmental reward signal is completely omitted, so how can HPO ensure that it is optimizing the original task rewards?

This definition also leads to an extreme case where the high-level policy simply generates the current state as the next subgoal, making the low-level policy do nothing and still "achieve" the subgoal. In this scenario, both the low-level policy and high-level policy would receive the highest reward, fully satisfying their optimization objectives. However, this would cause the agent's overall policy to just be idle. (My main concern is the problem formulation doesn't involve the MDP reward function).

Given this, I'm unclear how HPO is supposed to work.

[4] Kulkarni, Tejas D., et al. "Hierarchical deep reinforcement learning: Integrating temporal abstraction and intrinsic motivation." Advances in neural information processing systems (2016).

**Question 2**. Following up on Question 1, around Line 485, it is mentioned that "HPO consistently generates low average distance values, which implies that HPO mitigates non-stationarity." The subgoals generated by HPO are often "close" to the current state. Could this be due to the aforementioned definition of the high-level reward function, which evaluates whether the low-level policy has achieved the subgoal? If so, would generating only near subgoals prevent the agent from progressing toward the overall task objective?

**Question 3**. At around Line 347, could you further prove why the advantage equals the entropy of the policy ($A(s_t,g^*,g_t) = \beta log(\pi^H (g_t | s_t, g^*))$)? and how is the $\beta$ defined? The advantage directly equates to the entropy of the policy is not intuitive to me. Ziebart's paper studies a special case based on some assumptions, it may not be generally applicable to all RL problems.

I would like to increase the score if these concerns are addressed.

---

### Official Review · Reviewer_ZAGe · 2024-10-31

**Soundness:** 3
**Presentation:** 2
**Contribution:** 3
**Rating:** 6
**Confidence:** 4

**Summary:**

This paper formulate HRL as a bi-level optimization problem and transform it into a primitive-regularized DPO formulation. The proposed method HPO incoporates token-level DPO into Max-Ent RL for mitigating non-stationary issue and infeasible subgoal generation issue.

**Strengths:**

1. This paper proposes a primitive-regularized preference optimization approach for HRL, which is a novel try.
2. The derived DPO formulation has theoretical groundings.

**Weaknesses:**

HPO is sensitive to the two introduced hyperparameters, $\lambda$ and $\beta$, according to Figure 5 and Figure 6 in the Appendix. Further, it is not clear the values of $\lambda$ and $\beta$  used in each task of HPO in Figure 2-4.

**Questions:**

1. Based on the experimental settings detailed in the Appendix, the pick-and-place task appears simple enough for single-level RL methods to solve, as seen in environments like panda-gym [1], meta-world [2], and td-mpc [3]. Therefore, it is unclear what makes your experimental setup unique, given that none of the baselines aside from HPO achieve a satisfactory success rate.

   [1] Gallouédec, Quentin, et al. "panda-gym: Open-source goal-conditioned environments for robotic learning." arXiv preprint arXiv:2106.13687 (2021).

   [2] Yu, Tianhe, et al. "Meta-world: A benchmark and evaluation for multi-task and meta reinforcement learning." Conference on Robot Learning, PMLR, 2020.

   [3] Hansen, Nicklas, Xiaolong Wang, and Hao Su. "Temporal difference learning for model predictive control." arXiv preprint arXiv:2203.04955 (2022).

2. Could you clarify why HPO’s performance is relatively low on Maze navigation tasks?

3. To better illustrate HPO's ability to address non-stationarity and infeasible subgoal generation (Figure 3 and Figure 4), a comparison with the HAC baseline would be better, as HAC also addresses non-stationarity and is a recent work. I think this comparison should not pose too much burden on the authors, as HAC is already included as a baseline for success rate comparison in Figure 2.

4. There is a typo in line 761:  "in Figure 6."

**Details Of Ethics Concerns:**

Thank the reviewer Reviewer MsT6 for pointing out Submission5077.  I read both papers today and recognized that they indeed exhibit a high level of similarity, which raises concerns about the dual submission.

Key areas of overlap include the core idea (Eq.13 and Eq.15), and the problem solved. The diagrams and experimental settings outlined in both papers are strikingly similar. The graphics illustrate parallel structures for each algorithm’s architecture, with comparable layout and labeling, reinforcing the visual similarity. The written content of the related works, problem formulation, and technical approach sections shows marked resemblance in terminology and phrasing. Additionally, the cited references in the related works of each paper appear nearly in the same order.

Overall, the problem focus, methodology, and even application scenarios—complex robotic tasks like maze navigation and pick-and-place—are nearly identical. Thus I suggest they could be better presented as a single, comprehensive approach rather than separate works.

---

### Official Review · Reviewer_MsT6 · 2024-10-31

**Soundness:** 4
**Presentation:** 3
**Contribution:** 3
**Rating:** 8
**Confidence:** 4

**Summary:**

The authors present HPO, a hierarchical RL method that directly optimizes environment reward and preferences coming from sparse sub-goal reaching reward by a lower level policies. The specific objective is derived from DPO, and helps mitigate non-stationarity common to most HRL methods.

**Strengths:**

**Motivation:** Non-stationarity in HRL is a big issue and this paper presents a well-motivated solution for it.

**Comparisons:** The # of and relevancy of baselines is solid, this is a convincing set of comparisons.

**Experiments:** The experiments are performed on tasks well-suited for HRL, and the analysis on goal distance prediction against HIER and HAC demonstrates that HPO’s objective encourages sampling reachable goals for the lower-level policy.

**Clarity:** THe paper is overall quite clear and the walkthrough of how to obtain the objective was both interesting and easy to read.

**Weaknesses:**

**Clarity:** Overall clarity is good, but the reason *why* non-stationarity is solved should be explicated better, earlier in the paper. Non-stationarity occurs because a high level policy outputting a certain subgoal can result in a different reward later in training. The reason why this is solved is because the ***reward** for the high-level policy automatically adapts with the low level policies changin*g as it is based on the value function. The part I italicized isn’t that clearly presented in the paper.

- For example, when looking at Figure 1, it just looks like the Value function being given to DPO is the reason why non-stationarity is solved. The caption states “Since this preference-based learning approach does not depend on lower primitive, this mitigates non-stationarity. Note that since the current estimation of value function is used to regularize the higher policy, it does not cause non-stationarity.”
    - Instead, this can be simplified to some form of the italicized statement above; the current statement does not directly explain why.
- Similar comment for the introduction and after giving the full objective in Eq.14.

**Experiments:** Why not compare HAC and HIER on the same graphs in Figs 3/4? It’s a little strange to pick each one individually for a separate comparison when they can be compared on the same things.

**Minor Issues:**

- A high level policy discount factor is missing from Equations 4, 9, 10 and so on. Maybe it’s not necessary as the authors are considering the one-step DPO objective, but perhaps that could be mentioned?
- Figure 2 text size and line widths are too small

**Questions:**

From Eq. 6 to Eq. 7, the constraint that $V_{\pi_L} > V_{\pi_L^*}$ is dropped for $V_{\pi_L} > \delta$ due to the justification that for sparse-reward goal-reaching, $V_{\pi_L^*} > 0$ must be true. But this no longer optimizes the same objective, right? We still don’t know the ground truth value that $V_{\pi^*_L}$ should be; the writing seems to ignore this issue. A simple footnote or extra sentence of discussion stating this problem would make this part clearer.

**Details Of Ethics Concerns:**

Hi, I am also a reviewer for the following paper: https://openreview.net/forum?id=mJKhn7Ey4y&referrer=%5BReviewers%20Console%5D(%2Fgroup%3Fid%3DICLR.cc%2F2025%2FConference%2FReviewers%23assigned-submissions)

After reading both, I believe that the papers are so similar that they should not be two separate submissions. For example, the challenges they aim to solve in hierarchical RL are identical:

"The first (C1) is non-stationarity caused by evolving lower-level policies, which destabilizes the higher-level reward function (Chane-Sane et al., 2021). The second (C2) is the high-level policy’s tendency to generate subgoals that are infeasible for the lower-level policy to achieve."

and

"Limitation L1: non-stationarity due to evolving lower-level primitive policy, and Limitation L2: infeasible subgoal generation by higher-level policy(Chane-Sane et al., 2021). When the higher and lower level policies are trained concurrently in HRL, due to continuously changing and sub-optimal lower level policy, the higher level reward function and transition model become non-stationary."

from the first paragraph of the introduction for both.

Furthermore, their proposed solution is almost the same:

We propose a novel Hierarchical Preference Optimization (HPO) method that leverages primitiveregularized Direct Preference Optimization (DPO) to solve complex RL tasks using human preference data (Section 4). Our approach is principled; we derive it by reformulating the HRL problem as a bi-level optimization problem. To the best of our knowledge, this is the first work to utilize the bi-level optimization framework to develop a principled solution for HRL.

and

"The key idea underlying DIPPER is twofold: we introduce a DPO-based approach to directly learn higher-level policies from preferences, replacing the two-tier RLHF component in the scheme described above with a simpler, more efficient single-tier approach; we replace the reference policy inherent in DPO-based approaches, which is typically unavailable in complex robotics tasks, with a primitive-enabled reference policy derived from a novel bi-level optimization formulation of the HRL problem."

The resulting "novel" solution for the procedure proposed is in Eq 13 in this paper and Eq 15 in the other, and are exactly identical.

Finally, the experiment figures are almost identical, with near-identical performance between the two methods introduced in the two papers comparing against an identical set of baselines on an identical set of environments. This by itself isn't strange, but the near-identical performance points to how these methods are essentially the same. I believe it's the same authors too, as they cite the same paper (Singh et al 2024) as the main prior work they build upon, with the same art style for all figures.

The main differences between the two papers:

{0, 1} (this paper) reward for the low level policy vs {-1, 0} (the other paper) reward
The use of human preferences vs substituting them with environment-generated preferences.
I think this should be one paper, ablating the single choice of preferences vs environment-generated preferences. I don't believe they are sufficiently different to create two papers for.

---

> ### Author Response · Authors · 2024-11-13
> **Response regarding ethic concerns [part 1]**
>
> **General Response:** We sincerely thank the reviewer for taking the time to provide detailed feedback and for raising these concerns. We sincerely apologize for any misunderstanding caused by our writing, and we greatly appreciate the opportunity to clarify the differences between the two papers and address the reviewer’s comments in detail.
>
> > Comment 1: After reading both, I believe that the papers are so similar that they should not be two separate submissions. For example, the challenges they aim to solve in hierarchical RL are identical.
>
> **Response to Comment 1:** We acknowledge that both DIPPER and HPO papers share a similar motivation, as both aim to address challenges in hierarchical reinforcement learning (HRL) using ideas from preference optimization. However, we would like to highlight the key differences between the two. While both tackle non-stationarity and infeasible subgoal generation in HRL, the solution approach, derivations, and practical implementations are distinct. Below, we provide a detailed breakdown to address these concerns further.
>
> > **Comment 2:** The resulting "novel" solution for the procedure proposed is in Eq 13 in HPO paper and Eq 15 in DIPPER other, and are exactly identical.
>
>
> **Response to Comment 2:** We understand the concern about the similarity in the mathematical structure of the two equations. However, the underlying derivation, assumptions, and implementation of these equations are different. Below, we provide a side-by-side comparison of the two equations and highlight the key differences:
>
> Equation (15) from the DIPPER is given by:
>
> \begin{align}
> \mathcal{L}\^d =  - \mathbb{E}\_{(\tau^1, \tau^2) \sim \mathcal{D}}
> \bigg[&
> \log \sigma
> \bigg(
> \sum\_{t=0}^{T-1}
> \big(\alpha \log \pi\_U \big( g^1\_t \mid s^1\_t\big) - \alpha \log \pi\_U \big( g^2\_t \mid s^2\_t\big) + \lambda \underbrace{{(V\_{L}^{k}(s^1\_t, g^1\_t) -  V\_{L}^k(s^2\_t, g^2\_t) )}}\_{A:=}
> \big)
> \bigg)
> \bigg]. \tag{15}
> \end{align}
>
> Equation (13) from the HPO paper is given by:
>
> \begin{align}
> \mathcal{L}(\pi^H\_{\star}, \mathcal{D}) =  - \mathbb{E}\_{(\tau\^1, \tau\^2, y) \sim \mathcal{D}}
> \bigg[&
> \log \sigma
> \bigg(
> \sum\_{t=0}^{T-1}
> \big(\beta \log \pi\^H\_{\star} \big( g^1\_t \mid s^1\_t, g^{\star} \big) - \beta \log \pi^H\_{\star} \big( g^2\_t \mid s^2\_t, g^{\star} \big) + \lambda \underbrace{(V\_{\pi\_L}(s^1\_t, g^1\_t) -V\_{\pi\_L}(s^2\_t, g^2\_t))}\_{B:=}
> \big)
> \bigg)
> \bigg]. \tag{13}
> \end{align}
>
>
>
> **Key Differences between Eq. (15) and Eq. (13):** To highlight the difference between Eq. (13) and Eq. (15), let us consider the terms A in Eq. (15) and B in Eq. (13). We remark that the value function used in A is  $V_{L}^k$ is a k step approximation of the optimal value function (which is derived using huristics in DIPPER without rigorous mathematical justifications, but works in practice, and also requires higher value of $k$). This leads to a double-loop algorithm proposed in DIPPER. One loop is to obtain the value of $V_{L}^k$ (k iterations), and then another loop is to update the policy after calculating the gradient of Eq. (15).
>
> In contrast, let us consider the term B in Eq. (13) from HPO; we note that it just has the value function evaluations $V_{\pi_L}$ without any additional inner loop to calculate the optimal value function, which is required in Eq. (15). Therefore, the algorithm proposed in HPO is a single loop algorithm to solve the challenges of HRL.
>
> We concede that both approaches are trying to solve the challenges posed by HRL, but the solution approaches are different, which leads to different algorithms (two loop algorithm to solve Eq. (15) and only single loop algorithm to solve Eq. (13)). We agree with the reviewer that the contributions can be incremental, but we humbly request the reviewer not to raise an ethics flag because that was never the intentions and we extremely apologies if our writing has lead to the wrong impression.
>
>
> **Additional Fundamental Difference between DIPPER and HPO.**
>
> (i) DIPPER requires access to a reference policy $\pi_{ref}$ as mentioned in the objective in [Eq. (5), DIPPER]. Also the derivation and the loss function derived for DIPPER in Eq. (15) holds only for the specific design choice of the reference policy defined in [Eq. (9), DIPPER], which depends upon the optimal value function, which later requires a k-step approximation of the optimal value function.
>
> (ii) On the other hand, the algorithmic development in HPO is independent of any reference policy and does not require any such assumptions. The derivations in HPO are motivated by the developments in this paper (https://arxiv.org/pdf/2404.12358).

---

> > ### Author Response · Authors · 2024-11-13
> > **Response regarding ethic concerns [part 2]**
> >
> > **Remarks**: We deeply regret any confusion caused by our presentation and sincerely apologize if it gave the wrong impression. We humbly request the reviewer to reconsider raising an ethics flag, as our intention has never been to misrepresent contributions. We hope our clarifications have highlighted the distinctions. We are happy to add more experimental comparisons between both approaches.
> >
> > We greatly value your feedback and would be happy to provide further clarifications if necessary. Thank you for your time and consideration.

---

> ### Comment · Reviewer_MsT6 · 2024-11-18
>
> > Comparing EQ 15 and Eq 13, difference comes in $k$ step Value function
>
> This $k$ step value function is, for all practical purposes, a design choice. Ironically, Algorithm 1 of this paper, HPO, line 8, literally uses $V_\pi^k$ which is only in the DIPPER paper; this signifies essentially a direct copy-paste of the latex algorithm block from DIPPER to HPO's algorithm. I'm not convinced that $k>1$ is even relevant as the DIPPER paper does not list $k$'s value in the hyperparameter list.
>
> > Needing $\pi_{\text{ref}}$
>
> This reference policy is absorbed into the objective, and again, results in very little difference between the two algorithms.
>
> Thus I am unconvinced and will be keeping my ethics flag. If the ethics review sees nothing wrong, I will be reviewing both papers as standalone contributions. But, at least to me, submitting two nearly identical papers with the same claimed contributions and nearly identical algorithms/objectives that have nearly identical experimental results seems to be ethically flawed.
>
> I would suggest combining into one paper and resubmitting next time.

---

### Official Review · Reviewer_ELYQ · 2024-11-04

**Soundness:** 2
**Presentation:** 2
**Contribution:** 2
**Rating:** 5
**Confidence:** 2

**Summary:**

This paper is about (goal-conditioned) hierarchical reinforcement learning. The authors describe two key challenges in hierarchical reinforcement learning: training instability due to non-stationary of off-policy learning for the higher-level policy and generation of infeasible sub-goals by the higher-level policy. It proposes a hierarchical approach in which the higher-level policy is optimized with a token-level direct preference optimization method and the lower-level policy is optimized with reinforcement learning. The goal of this approach is to make the learning of the higher-level policy independent from the lower-level policy (i.e. its current sub-optimal form) to avoid issues arising from non-stationarity. To this end, the paper re-formulates the hierarchical reinforcement learning problem as a bi-level optimization problem which is solved by first posing an equivalent constrained optimization problem. The proposed method is evaluated in a set of experiments and compared to a set of baselines.

**Strengths:**

The paper provides good background on reinforcement learning from human feedback and direct preference optimization. The paper also clearly describes the limitations it aims to address. The authors introduce a bi-level formulation of the hierarchical reinforcement learning problem to provide formalized arguments for the issues that they want to address. The overall issue that is raised in this paper, i.e. the complications arising for the interplay between the high-level and low-level policies is highly relevant for hierarchical reinforcement learning and satisfying solutions for this problem are in demand.

**Weaknesses:**

In parts, this paper is hard to follow. For example, the part where the notation and the sub-goals are introduced is confusing as to the nature and purpose of the sub-goals. More clarity as to the definition of the hierarchical MDP would be good. Another reason is the level to which the paper is self-contained. For example, in line 206, the authors use an equation for the optimal policy with reference to a tutorial, but it is unclear what the equation means and why it is used.

**Questions:**

In Fig. 3, the authors are presenting a form of evaluation for their claim regarding non-stationarity. This evaluation is indirect and is relying on the task (i.e. what distances mean). Can the authors present a task independent evaluation that supports their claim? E.g. sometime closer to the formalization?

---

### Note · Authors · 2024-11-18

I have read and agree with the venue's withdrawal policy on behalf of myself and my co-authors.